# Antifungal Capacity of Microcapsules Containing *Lactiplantibacillus plantarum* TEP15 or *Lactiplantibacillus pentosus* TEJ4

Yeni Pérez-Ruiz, David Mejía-Reyes, Raymundo Rosas-Quijano, Didiana Gálvez-López, Miguel Salvador-Figueroa and Alfredo Vázquez-Ovando *

Instituto de Biociencias, Universidad Autónoma de Chiapas, Boulevard Príncipe Akishino sin número Colonia Solidaridad 2000, Tapachula 30798, Chiapas, Mexico; yeni.perez44@unach.mx (Y.P.-R.); jose.mejia@unach.mx (D.M.-R.); raymundo.rosas@unach.mx (R.R.-Q.); didiana.galvez@unach.mx (D.G.-L.); miguel.salvador@unach.mx (M.S.-F.)
* Correspondence: jose.vazquez@unach.mx; Tel.: +52-9626427972

**Abstract:** The use of lactic acid bacteria (LAB) for the biocontrol of fungi that cause fruit and vegetable deterioration is a highly promising strategy. However, one of the main challenges lies in maintaining both viability and antifungal activity even in conditions that are unfavorable for LAB. The microencapsulation of LAB can minimize the impact of environmental conditions, helping to maintain viability, but there is still little information on what occurs with respect to antifungal activity. Therefore, the objective of this study was to evaluate the effect of microencapsulation with several polymer blends on the viability and antifungal activity of *Lactiplantibacillus plantarum* TEP15 and *L. pentosus* TEJ4. Sodium alginate (2%), sodium alginate–gum arabic (2%:1%), sodium alginate–k-carrageenan (2%:0.05%), and sodium alginate–whey protein (0.75%:1.5%) were used as encapsulating polymers. After processing (day 0), as well as after 14 and 28 days of storage, the encapsulation efficiency, moisture content, bead size, and survival were evaluated. Likewise, the encapsulated bacteria were subjected to in vitro tests against *Colletotrichum gloeosporioides*, *Penicillium* AT21H10, and *Fusarium* sp. Capsules with sizes ranging from 1.47 mm to 1.88 mm were obtained, and all the wall materials tested had more than 85% encapsulation efficiency and allowed survival rates greater than 90% and 80% to be maintained after 14 and 28 days of storage, respectively. The encapsulated LAB inhibited the growth of mycelia by up to 100%, but, against spores, the greatest inhibition was 22.08% for all the fungi tested.

**Keywords:** biocontrol; extrusion; gum arabic; lactic acid bacteria; phytopathogenic fungi; sodium alginate; whey protein

## 1. Introduction

The deterioration of fruits and vegetables caused by the presence of fungi causes significant economic losses, especially in tropical regions around the world. It is estimated that approximately 5–10% of global food is lost each year in the tropics due to fungal spoilage [1]. Synthetic fungicides are the most commonly used compounds to control these pathogens; however, the excessive use of these products has caused undesirable effects, such as environmental contamination, the emergence of fungicide resistance in pathogens, and damage to human and animal health [2]. Consequently, new alternatives have been sought to the use of synthetic products, including physical methods (low temperatures, high hydrostatic pressure, pulsed electric fields, cold plasma, and radiation), phytoextracts (plant extracts, rotenone, and essential oils, among others), and, more recently, biological alternatives such as biocontrol [3,4].

Biocontrol is conceived as a strategy that uses microorganisms or their metabolites to inhibit the growth and multiplication of pathogens [5]. Among the microorganisms that can be used as biocontrol agents for fungi or to produce natural antifungal agents are lactic acid bacteria (LAB) [6]. It has been reported that various LAB species can inhibit the growth of various phytopathogenic fungi [7–9], including the most common ones, namely, *Fusarium* spp. (causal agent of Fusarium disease), *Colletotrichum gloesporioides* (causes anthracnose in a wide range of fruits), and *Penicillium* spp. (accelerates the decomposition process in citrus fruits during harvest and postharvest) [10]. The inhibitory effect that LAB have on these fungi is achieved through different mechanisms, the most efficient being the production of metabolites, which include organic acids, antimicrobial peptides, hydrogen peroxide, alcohols, fatty acids, diacetyl, reuterin, and bacteriocins, among others [11–13].

Although some LAB are very efficient at controlling fungi, these microorganisms are generally sensitive to environmental conditions, such as temperature, air, humidity, and pH [1,14]. However, as microorganisms classified as GRAS (generally recognized as safe) by the FDA, they are considered safe for both humans and the environment, which makes them ideal candidates for various applications, including biocontrol [6,15]. Therefore, alternatives have been sought to reduce the loss of viability of LAB in the face of hostile environmental conditions, highlighting encapsulation as an option for this purpose. This technique seeks to use a protective barrier to protect LAB from adverse factors in their environment [16].

Various encapsulation methods have been proposed in recent years, such as emulsification, coacervation, and spray drying, among others [17]. However, some methods have certain disadvantages, such as their high cost or the requirement for infrastructure. In contrast, the extrusion technique has been presented as an alternative to these disadvantages because this technique can reduce cell damage, increase encapsulation efficiency, and maintain the antifungal capacity of the encapsulated bacteria [18]. Sodium alginate has been reported as the most common coating material for extrusion encapsulation due to its simplicity, zero toxicity, biocompatibility, and low cost [19,20]. This polymer has been used either as the sole encapsulating agent [21] or in combination with other polymers, such as polysaccharides (carrageenan), gums (gum arabic), lipids, or proteins (whey protein), to improve the mechanical and chemical stability of microcapsules [14,22].

In addition to maintaining viability, encapsulation has positive effects on the antifungal activity of the encapsulated bacteria [23,24]. We previously isolated two LAB strains, *Lactiplantibacillus plantarum* TEP15 and *Lactiplantibacillus pentosus* TEJ4, which inhibit the development of *Colletotrichum gloeosporioides* [25]; however, when applied in situ, these strains exhibit reduced antifungal activity. Therefore, the objective of the present work was to evaluate the effect of microencapsulation on the antifungal activity of *Lactiplantibacillus plantarum* TEP15 and *L. pentosus* TEJ4 using different combinations of encapsulating polymers. The hypothesis of this work states that at least one of these combinations can maintain the antifungal activity of at least one strain.

## 2. Materials and Methods

### 2.1. Microorganisms

The lactic acid bacteria *Lactiplantibacillus plantarum* TEP15 and *Lactiplantibacillus pentosus* TEJ4 were previously isolated from fermented beverages [tepache (TEP) and tejuino (TEJ)] [24]. Three strains of phytopathogenic fungi were also used: *Colletotrichum gloeosporioides* (isolated from Maradol papaya), *Penicillium* AT21H10 (isolated from banana), and *Fusarium* sp. (isolated from papaya). All biological material was provided by the strain collection of the Instituto de Biociencias of the Universidad Autónoma de Chiapas, Mexico.

#### 2.1.1. Lactic Acid Bacteria

The two LAB strains were reactivated by continuous reseeding in Man, Rogosa and Sharpe (MRS) (Difco™, Detroit, MI, USA) agar and broth at a pH of 6.5 for 48 h, and their morphological characteristics were verified by optical microscopy, and their biochemical

characteristics were determined by Gram staining and catalase tests. Subsequently, growth kinetics were carried out for each strain; for this, a roast of the microorganism was taken, cultured in 5 mL of MRS broth, and stirred at 150 rpm in an orbital shaker (Luzern® H2-300, Lucerne, Switzerland) for 48 h. Afterwards, the inoculum was added to a flask with 100 mL of MRS broth and incubated under the conditions mentioned above. Aliquots were taken at 12, 24, 36, and 48 h, and the absorbances were read by spectrophotometry at 560 nm (Thermo Scientific Genesys™ 20 model 4001/4, Waltham, MA, USA) to determine the time at which the highest optical density was reached. Once this value was reached (36 h), an aliquot was taken and subjected to serial dilutions, and a plate count was performed to estimate the number of CFU $mL^{-1}$.

### 2.1.2. Phytopathogenic Fungi

Mycelium discs from the fungi *C. gloeosporioides*, *Penicillium* AT21H10, and *Fusarium* sp. were reseeded in Petri dishes with potato dextrose agar (PDA) (MCD-LAB, Oaxaca, Mexico) at a pH of 7 and incubated for 7 days until the mycelia covered the total area of the plates.

PDA culture medium was prepared in Roux bottles, and mycelium discs 5 mm in diameter from the plate culture were reseeded and incubated until sporulation of the fungi was observed. The spores were collected by washing with Ringer's solution and sterile glass beads. Aliquots of each solution were placed in a Neubauer chamber, the spores were quantified microscopically at a magnification of 40×, and the concentration of the solutions was adjusted to $1 \times 10^6$ spores $mL^{-1}$.

### 2.2. Microencapsulation

To investigate the effect of the microcapsule formulation on the response variables, a completely randomized experiment with a $2 \times 4$ factorial arrangement was carried out. Two strains of lactic acid bacteria and four formulations were tested, which had been previously developed in other studies [26–29], for a total of 8 treatments (Table 1).

**Table 1.** Combinations of two strains of lactic acid bacteria and four formulations (sodium alginate alone or combined with gum arabic, κ-carrageenan, or whey protein) through a factorial design.

| Treatment Code | Strain | Formulations |
| --- | --- | --- |
| ALG15 | *L. plantarum* TEP15 | Sodium alginate (2% *w/v*) |
| ALGWPC15 | *L. plantarum* TEP15 | Sodium alginate (0.75% *w/v*) + whey protein (1.5% *w/v*) |
| ALGCAR15 | *L. plantarum* TEP15 | Sodium alginate (2% *w/v*) + κ-carrageenan (0.05% *w/v*) |
| ALGGA15 | *L. plantarum* TEP15 | Sodium alginate (2% *w/v*) + gum arabic (1% *w/v*) |
| ALG4 | *L. pentosus* TEJ4 | Sodium alginate (2% *w/v*) |
| ALGWPC4 | *L. pentosus* TEJ4 | Sodium alginate (0.75% *w/v*) + whey protein (1.5% *w/v*) |
| ALGWPC4 | *L. pentosus* TEJ4 | Sodium alginate (2% *w/v*) + κ-carrageenan (0.05% *w/v*) |
| ALGGA4 | *L. pentosus* TEJ4 | Sodium alginate (2% *w/v*) + gum arabic (1% *w/v*) |

Briefly, each LAB strain was inoculated in 100 mL of MRS broth and agitated for 36 h. The biomass of the cell culture was obtained following the methodology reported by Praepanitchai et al. [30]. To achieve this, the bacterial suspension was centrifuged at $3381 \times g$ for 10 min at 4 °C (Beckman Coulter centrifuge model Allegra™ 64R, Fullerton, CA, USA), and the cell pellet was washed with 0.9% *w/v* NaCl (Binden Reagents, Ecatepec de Morelos, Mexico) and resuspended in 250 μL of saline solution for later use.

To prepare the microcapsules by extrusion, the procedure described by Parsana et al. [31], with slight modifications in cell volume, needle caliber, and polymer concentration, was followed. In triplicate, 100 mL of the encapsulating solution was prepared and mixed with the cell biomass previously obtained by centrifugation. The mixture was gently shaken and placed into a syringe with a 0.6 mm needle (31 G, BD Ultra-Fine™, Becton Dickinson, Franklin Lakes, NJ, USA). Then, this mixture was extruded dropwise against a sterile

0.1 M $CaCl_2$ solution under stirring at 150 rpm for 30 min and filtered with Whatman No. 4 filter paper. The microcapsules were washed with saline (0.9% $w/v$) and stored in airtight containers with peptone water (0.1% $w/v$) at 4 °C.

## 2.3. Encapsulation Efficiency

The encapsulation efficiency (EE %) was determined by counting live cells through serial dilutions following the procedures described by Castro et al. [32] and Petraitytė et al. [33]. Under aseptic conditions, one g of the microcapsules (in triplicate) was weighed and added to 9 mL of 1 N sodium citrate solution. This sample was stirred for 10 min using a vortex shaker (Vortex Cole-Parmer®, Antylia Scientific, Vernon Hills, IL, USA). Subsequently, the sample was subjected to serial dilutions, transferring 1 mL of the previous suspension to a tube with 9 mL of peptone water. Finally, 1 mL of the −6, −7, and −8 dilutions was deposited in Petri dishes containing MRS agar and plated using the pour-plate technique. The plates were incubated at room temperature for 36 h under anaerobic conditions. After this time, the colonies were enumerated, and the results were expressed in CFU $g^{-1}$. The encapsulation efficiency (EE) was calculated using Equation (1):

$$EE\ (\%) = N/No \times 100 \tag{1}$$

where N is the log number of viable cells (CFU) released from the microcapsules, and No is the log number of cells free viable cells (CFU) added to the polymer mixture during the production of microcapsules.

## 2.4. Microcapsule Characterization

### 2.4.1. Moisture Content

The measurement of the moisture content of the microcapsules was based on the difference in weight between the wet sample and the dry sample, following the technique reported by Avila-Reyes et al. [34]. In triplicate, in crucibles previously set to a constant weight (crucible weight), one g of the sample was weighed in triplicate and placed in a Felisa® oven (Jalisco, México) for 3 h at 70 °C. After this time, the crucibles were removed from the oven and cooled for 15 min in a desiccator to finally be weighed (final weight). The percentage of humidity was obtained using Equation (2):

$$moisture\ (\%) = [sample\ weight - (final\ weight - crucible\ weight)/sample\ weight] \times 100 \tag{2}$$

### 2.4.2. Microcapsule Size

The microcapsules were measured with a Carl Zeiss AxioLab 1 model optical microscope equipped with an AxioCam ERc5s digital camera using the Zen™ program. The diameter of 10 microcapsules was randomly measured with a 40× objective. The capsule size was expressed in millimeters (mm).

## 2.5. Storage for 28 Days

The microcapsules containing the LAB were stored (in triplicate) in peptone water (0.1%) for 28 days at 4 °C. During storage, the survival rate and moisture content were measured at 14 and 28 days. Moisture was estimated following the procedure described above. The survival rate was determined by counting live cells by means of serial dilutions, following the procedure described above, and calculated using Equation (1), where No (CFU $g^{-1}$) is the number of viable cells before storage, and Nt (CFU $g^{-1}$) is the number of viable cells at the end of storage.

## 2.6. In Vitro Antifungal Activity of Microencapsulated Bacteria against Colletotrichum gloeosporioides, Fusarium sp., and Penicillium AT21H10

The antifungal capacity of the microcapsules against the mycelia and spores of the three phytopathogenic fungi was evaluated. The confrontation was carried out with freshly prepared capsules as well as after 14 and 28 days of storage.

### 2.6.1. Evaluation against Mycelium

For confrontations against fungal mycelium, the dual-culture technique described by Michel-Aceves et al. [35] was used with some modifications. In quintuplicate, Petri dishes containing PDA agar were prepared, an 8 mm diameter disc with mycelium of the previously described fungi was deposited on one end of the plate, and, on the opposite side, an 8 mm diameter well was made in the agar. Then, 200 mg of microcapsules was placed in the well. The plates were incubated for 10–12 days until the control plate (mycelium plated on ADP agar without microcapsules) covered the total area of the plate. The growth area of the fungus was calculated from the diameter measured with a Vernier caliper. The percentage of inhibition (%) was calculated using Equation (3):

$$I\ (\%) = [(Ac - At)/Ac] \times 100 \tag{3}$$

where Ac is the total area of the control plate, and At is the growth area of the pathogen in the confrontation.

### 2.6.2. Evaluation against Spores

To verify the capacity of the encapsulated LAB strains to inhibit the germination of spores of phytopathogenic fungi, the agar well diffusion method reported by Mohammadi et al. [1] was followed with some modifications. In quintuplicate, eight mL of PDA agar that had been previously inoculated with 100 µL of the fungal spore suspension ($1 \times 10^6$ spores mL$^{-1}$) was poured into Petri dishes and allowed to solidify. Subsequently, 200 mg of the encapsulated LAB was added to each 8 mm diameter well. The plates were incubated at 30 °C for 4 days until the control plate (containing spores without microcapsules) covered the total area of the plate. The inhibition of spore germination and fungal growth was determined by measuring the area of inhibition surrounding each agar well, and the result was calculated using Equation (3), where Ac is the total area of the control plate, and At is the growth area of the pathogen in the confrontation.

### 2.7. Data Analysis

The data obtained were subjected to an analysis of variance (ANOVA), and a comparison of means was subsequently performed using Tukey's test ($p < 0.05$) with the InfoStat software version 2020.

## 3. Results and Discussion

The results of the growth kinetics of both LAB strains demonstrated that, at 36 h, the highest optical density was obtained (1.810 absorbance units for the TEP15 strain and 2.242 absorbance units for the TEJ4 strain), equivalent to a cell concentration of $2.468 \times 10^9$ CFU mL$^{-1}$ and $3.384 \times 10^9$ CFU mL$^{-1}$, respectively.

### 3.1. Encapsulation Efficiency

The encapsulation efficiency (EE) ranged from 86 to 94% (Figure 1), with significant differences ($p < 0.05$) between treatments. The highest values were found in the treatments in which sodium alginate was used as the only encapsulating agent and in those where alginate was combined with whey protein, with the ALG4 treatment having the highest average value. In contrast, the treatments with the lowest efficiency were those in which alginate with κ-carrageenan and alginate with gum arabic were used. Furthermore, the results indicated that, on average, there was a greater encapsulation efficiency of *L. pentosus* TEJ 4 than of *L. plantarum* TEP15, which can be attributed to the characteristics of the strain [36,37]. However, this finding contrasts with what was reported by Bagdat et al. [38], who encapsulated three probiotic strains, resulting in a greater encapsulation efficiency for *L. plantarum* subspecies W2, with values ranging between 87% and 98.36%, than those obtained for *Limosilactobacillus fermentum* W8 (61.87–98.36%) and *L. pentosus* XL640 (71.66–95.37%). Certain strains of LAB can tolerate and adapt to various stress conditions [39], which can increase their ability to maintain cell viability during the microencapsulation process.

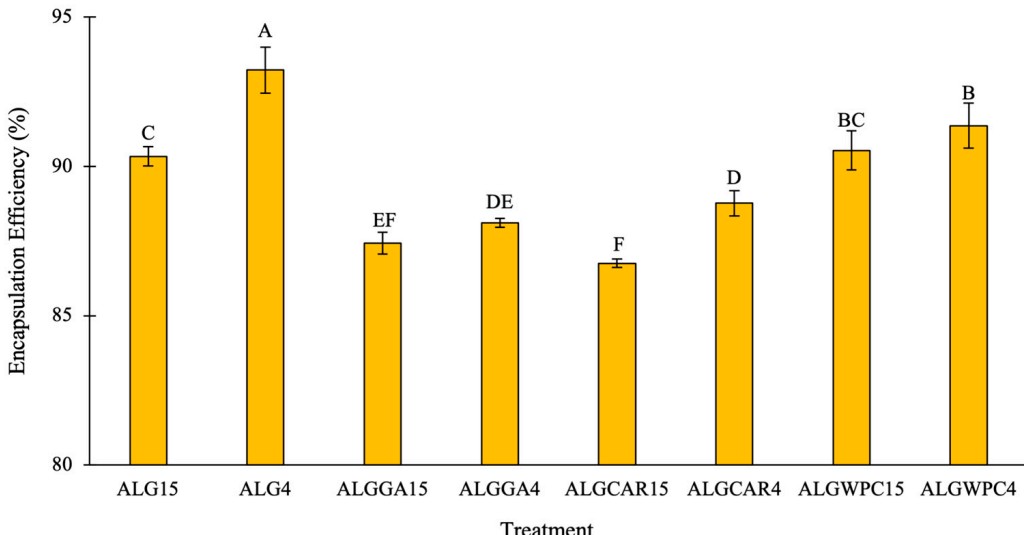

**Figure 1.** Encapsulation efficiency of LAB in different microcapsules formulations. Different letters indicate significant differences ($p < 0.05$) according to Tukey's test. The lines above the bars show the standard deviation. For details of the codes used for the treatments, see Table 1.

The high percentages of encapsulation efficiency obtained in the present study correlate strongly with the technique used [14,40]. Extrusion is a method that involves mild operating conditions that reduce cell damage and, therefore, guarantee high levels of viable cells during encapsulation [18]. In addition to the technique used, the encapsulation efficiency is also affected by the wall materials used [41]. Castro-Rosas et al. [32] encapsulated *L. paracasei*, obtaining results slightly lower than those in the present work, with an efficiency of 87.6%, using only alginate; this may be because sodium alginate is an effective encapsulating polymer in terms of cell trapping [41]. However, when different polymers are combined, the encapsulation efficiency can be affected by the interactions between the materials that make up the capsule wall, as occurred in the treatments in which alginate had been combined with gum arabic or κ-carrageenan, which presented lower efficiencies than the treatments in which only sodium alginate had been used (Figure 1). Sandoval-Mosqueda et al. [27] obtained a slightly lower efficiency than this study (84.71%) using alginate with gum arabic. This is because gum arabic can have a negative effect on some microorganisms, as it acts as a barrier around the bacterial cell wall, affecting its permeability and osmotic balance, causing water filtration and the release of cell components to the outside [27,42]. In contrast, Batalha et al. [28] reported that encapsulation using κ-carrageenan at a concentration greater than 1% in combination with alginate results in efficiencies greater than 90%, which can be attributed to the fact that a higher concentration of κ-carrageenan improves the structure and stability of the encapsulated microorganisms [29]. The combination of alginate with whey protein achieved an EE above 90%, which can be attributed to the fact that whey protein is considered a suitable wall material for encapsulation since, when combined with other polymers, whey protein provides a protective layer which protects the bacterial cell against damage to the cell wall, which increases its viability and encapsulation efficiency [43,44].

### 3.2. Moisture Content

The results of the moisture content of the freshly prepared beads as well as after 14 and 28 days of storage at 4 °C are presented in Table 2. Although significant differences ($p < 0.05$) were found between the treatments, the values were very similar for each storage time, thus showing little influence of the encapsulating agents. In contrast, the moisture content gradually increased as time passed, demonstrating that the beads of six of the eight treatments allowed the entry or incorporation of water into the encapsulating matrix from the outside. Only the capsules made with alginate and gum arabic did not show

changes in their moisture content ($p > 0.05$) with respect to storage time. The moisture content is an important factor influencing the viability of microencapsulated bacteria. Some reports suggest that a high moisture content can have a negative impact on the stability and effectiveness of encapsulated bacteria [45] and that lower moisture values allow the viability of LAB to be preserved during storage because a reduced moisture content can slow down their metabolic processes and increase their stability [46,47]. However, even with the values obtained in this work, moisture had no influence on the viability of the encapsulated bacteria. The moisture values of the microcapsules obtained in our study are similar to those obtained by Fathi et al. [14], who reported moisture contents between 78 and 97%, and by Poletto et al. [48], who reported moisture percentages ranging from 81.64 to 94.94%.

**Table 2.** Moisture content (%) of freshly made microcapsules and after 14 and 28 days of storage at 4 °C.

| Treatment | Day 0 | | Day 14 | | Day 28 | |
|---|---|---|---|---|---|---|
| ALG15 | 94.57 ± 1.21 | ab B | 95.87 ± 1.50 | a AB | 97.24 ± 0.31 | a A |
| ALG4 | 93.71 ± 1.31 | b B | 94.45 ± 0.75 | b B | 95.96 ± 0.54 | d A |
| ALGGA15 | 95.31 ± 1.34 | ab A | 96.67 ± 0.34 | a A | 96.88 ± 0.18 | abc A |
| ALGGA4 | 95.77 ± 1.10 | a A | 96.11 ± 0.80 | a A | 96.93 ± 0.18 | ab A |
| ALGCAR15 | 94.80 ± 0.12 | ab C | 95.84 ± 0.66 | a B | 97.10 ± 0.18 | a A |
| ALGCAR4 | 95.21 ± 0.35 | ab B | 95.97 ± 0.13 | a A | 96.26 ± 0.25 | cd A |
| ALGWPC15 | 95.13 ± 0.20 | ab C | 95.80 ± 0.60 | ab B | 96.42 ± 0.20 | bcd A |
| ALGWPC4 | 94.59 ± 0.41 | ab B | 95.43 ± 0.45 | ab A | 95.81 ± 0.27 | d A |

The data are presented as the average ± standard deviation. Values with different letters indicate significant differences ($p < 0.05$) according to Tukey's test. Lowercase letters denote differences between treatments for the same storage time. The capital letters denote differences between storage times for the same treatment.

### 3.3. Microcapsule Size

The average size of the microcapsules varied from 1.47 mm to 1.88 mm (Figure 2). The sizes of the beads of alginate and alginate with κ-carrageenan were significantly equal ($p > 0.05$) to each other and different ($p < 0.05$) with respect to the treatments in which alginate with gum arabic and alginate with whey protein had been used. The microcapsules from the ALGGA15 treatment had the smallest average size, while the largest bead size was obtained from the ALG4 treatment. Although the sizes of the beads in all the treatments were within a narrow range, the significant differences could be attributed to interactions that occurred between the polymers rather than to the amount of dry matter in the formulation. That is, although there were treatments with a greater amount of dry matter (sodium alginate + gum arabic), these were not the greatest. It can be thought that sodium alginate, when interacting with both κ-carrageenan and whey, forms more compact networks that reduce the size of the capsule. The electrostatic interaction between the amino group of whey protein and the carboxyl group of alginate can support this behavior [14]. The opposite could occur in the interaction of sodium alginate with gum arabic, which does not modify the size of the capsules made from only sodium alginate [49]. As expected, due to the nature of the extruder used, the size of the capsules was in the range of mm. It has been reported that the main drawback of using the extrusion technique is the slow formation and larger size of the microcapsules [40], which can even reach diameters of up to 5 mm [50]. The results for the size of the microcapsules are similar to those reported by Fareez et al. [51], who, as in the present study, used a 0.6 mm diameter needle to produce capsules with sizes ranging from 1.312 to 1.343 mm. However, Sandoval-Mosqueda et al. [27], Luca et al. [52], and Ta et al. [53] reported the formation of larger capsules (1.7–2.0 mm, 1.86–2.25 mm, and 2–5.5 mm, respectively). The great variability in the size of the microcapsules obtained by extrusion may be due

to different factors, such as the viscosity and concentration of the polymer solution, the needle–solution distance, the stirring speed of the hardening solution, and the diameter of the needle [18,54].

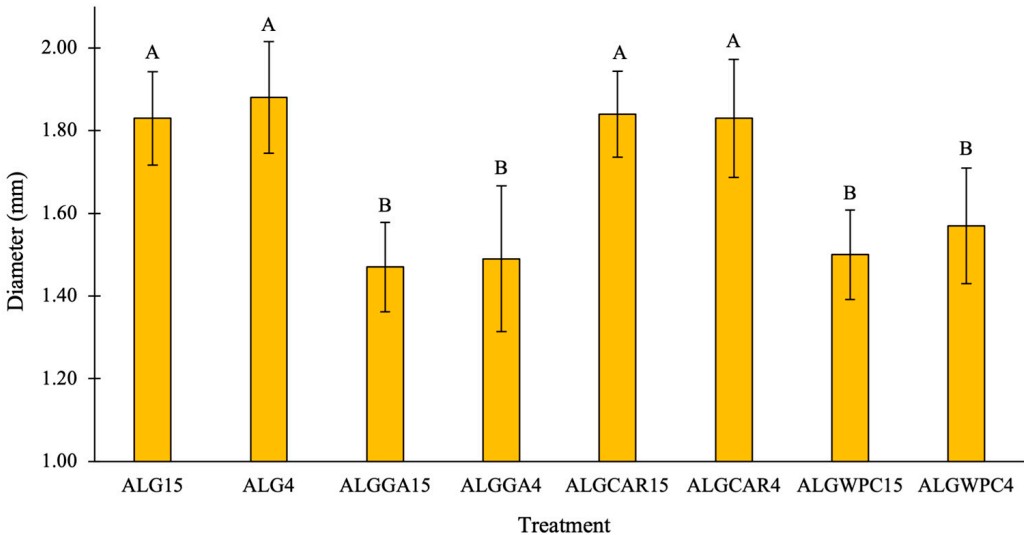

**Figure 2.** Microcapsule size. Different letters indicate significant differences ($p < 0.05$) according to Tukey's test. The lines above the bars show the standard deviation. For details of the codes used for the treatments, see Table 1.

### 3.4. Survival Rate

The viability of the microencapsulated bacteria after 14 and 28 days of storage at 4 °C is shown in Figure 3. On average, the survival rate of the treatments in which sodium alginate had been combined with other polymers was greater than that of the treatments in which only alginate had been used. This may be because sodium alginate capsules tend to be porous to cellular material, which makes them susceptible to releasing their contents or allowing the entry of substances which impair viability [20,55–57]. After 14 days of storage, the combination of alginate with κ-carrageenan resulted in the highest survival values, which were 97.72% and 98.72% when the TEP15 and TEJ4 strains were encapsulated, respectively. Azam et al. [29] mentioned that the gelling properties of κ-carrageenan can cause increased viscosity, resulting in a slow release of the encapsulated material. However, after 28 days of storage, these capsules showed a noticeable decrease in survival. This phenomenon may be because capsules made with κ-carrageenan may be fragile and not able to withstand the stresses of internal bacterial growth beyond 14 days [16,58] or because the material loses its stability after 14 days and releases the encapsulated material. At the end of the storage period (day 28), the capsules made with alginate and whey protein had the greatest viability, possibly because, as a wall material, the combination of whey protein and sodium alginate provided greater protection to the bacteria due to their ability to form a stable encapsulating matrix without compromising their survival. The type of intermolecular interaction may be responsible for this behavior, such as the previously mentioned electrostatic interactions between alginate and whey protein [14]. In addition, sodium alginate provides a protective structure, while whey protein provides essential nutrients, helping to maintain ideal conditions for bacterial survival during long-term storage [59,60]. The results obtained in our study are superior to those reported by Castro et al. [32], who reported an 89.2% survival of *Lactobacillus paracasei* microencapsulated and stored at 4 °C for 6 weeks.

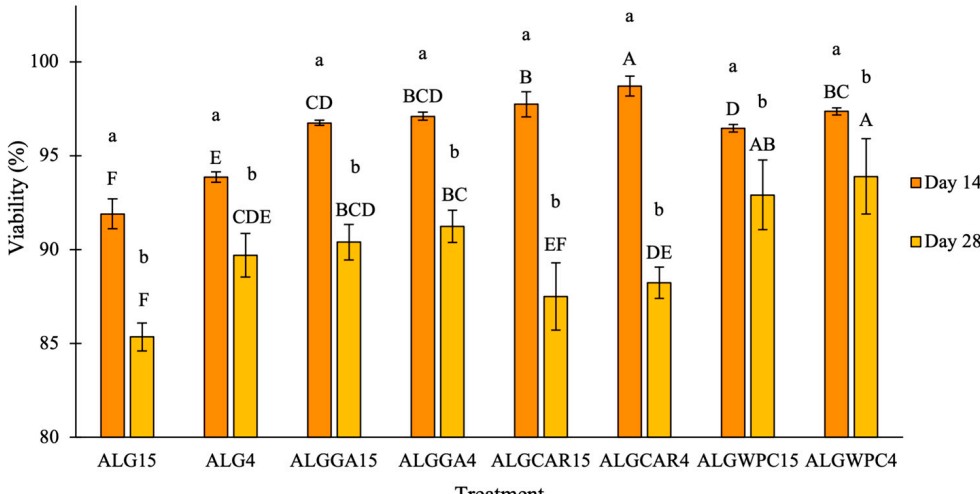

**Figure 3.** Viability of microencapsulated LAB after 14 and 28 days of storage at 4 °C. Different letters in bars of the same color indicate significant differences ($p < 0.05$) according to Tukey's test. The capital letters denote comparisons between treatments for the same date (bars of the same color). Lowercase letters denote comparisons between storage times for the same treatment. The lines above the bars show the standard deviation. For details of the codes used for the treatments, see Table 1.

### 3.5. In Vitro Antifungal Activity against the Mycelium of Fungi

The results of the antifungal activity of the capsules containing freshly prepared LAB and during storage against the three phytopathogenic fungi showed a particular behavior for each treatment (Figure 4). In some treatments, an increase in the inhibition of fungal growth was observed at 14 days compared to day 0. However, this behavior was not maintained for 28 days, since these microcapsules reduced their antifungal capacity, as occurred in the ALG15 treatment. In contrast, for the capsules of the ALG4 and ALGWPC4 treatments, the opposite pattern of behavior occurred, since there was a decrease in inhibition at 14 days compared to the initial measurement on day 0. However, at 28 days, there was an increase in antifungal inhibition. Additionally, in another group of treatments, there was variation; however, the inhibitory capacity was approximately the same during the 28 days, as could be observed in the ALGWPC15 treatment. An explanation for this variability in antifungal activity could be related to the dynamics of release of the encapsulated material (cells or postbiotics) and, in turn, to the components that make up the capsule wall, since, depending on the encapsulating agents, the release rate of the encapsulated material will be different [34,61]. For the LAB used here, it has been shown that postbiotics (evaluated as cell-free supernatants, CFSs) produce a greater-than-80% inhibition against *C. gloeosporioides*, *Fusarium* sp., and *Penicillium* AT21H10 [10]. The above findings suggest that the treatments in which whey protein with alginate had been used as the encapsulating material did not allow the release of the content inside the capsules to a large extent, since, even though they were the treatments that had greater viability at the end of the storage period (Figure 3), the same storage day had the lowest inhibition values (Figure 4). Additionally, the survival of the bacteria within the capsule must be considered, which depends on the permeability of the matrix for the supply of nutrients and the elimination of toxic metabolites which directly influence their survival or cell death [62,63]. A decrease in the number of bacteria could cause a decrease in the synthesis of antimicrobial compounds [64], as occurred in the treatments in which carrageenan had been used, since, on day 28, a decrease in viability was observed compared to day 14 (Figure 3), which could cause the antifungal capacity of these bacteria to decrease (Figure 4). Other research has reported higher values of fungal inhibition of encapsulated microorganisms. Rubio-Tinajero et al. [23] evaluated the in vitro antagonistic effect of alginate microcapsules containing *Trichoderma* spp. and *Bacillus* spp., which inhibited *Fusarium oxysporum* by 84.7% and 83.7%, respectively.

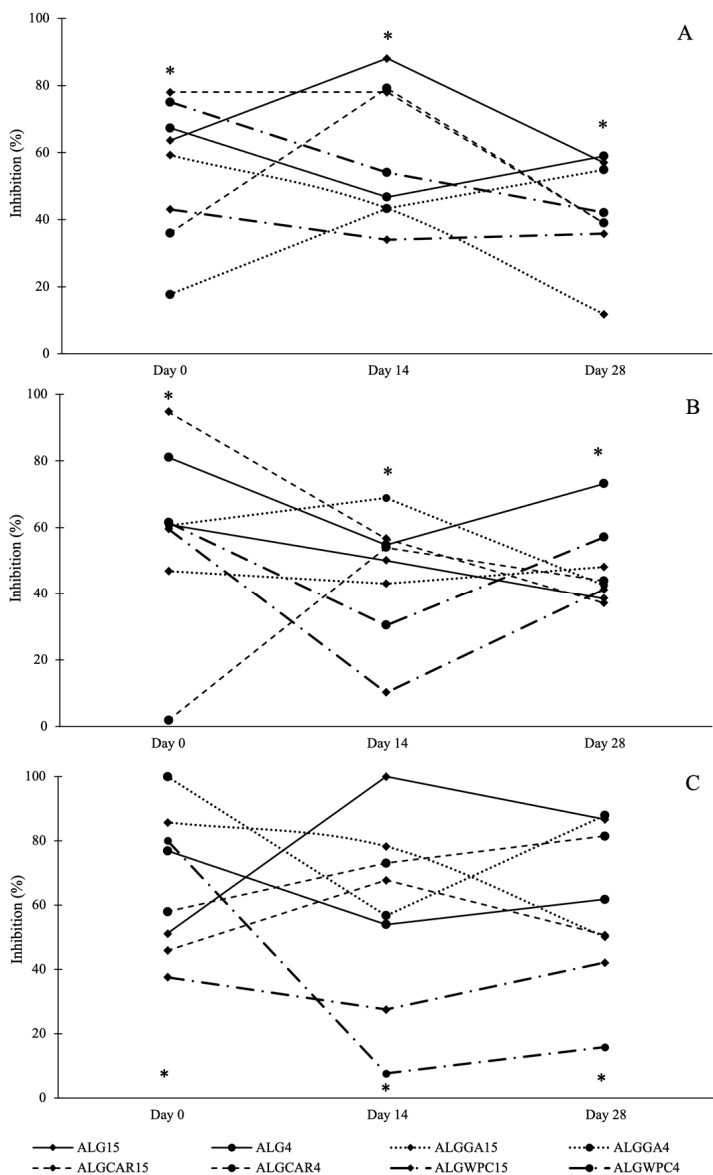

**Figure 4.** Inhibitory effects of microencapsulated LAB on the mycelia of *Colletotrichum gloeosporioides* (**A**), *Fusarium* sp. (**B**), and *Penicillium* AT21H10 (**C**). Asterisks indicate the days where there were significant differences between the treatments ($p < 0.05$). For details of the codes used for the treatments, see Table 1.

## 3.6. In Vitro Antifungal Activity against Spore Germination

The spore germination inhibition values of phytopathogenic fungi with freshly prepared microcapsules (day 0) and after 14 days of storage at 4 °C are shown in Figure 5. Because the microcapsules after 14 days of storage presented low and lower inhibition values compared to their counterparts on day 0, evaluations of the capsules stored for 28 days were not carried out. It was observed that the freshly prepared microcapsules (day 0) inhibited the growth of *Fusarium* sp. to a greater extent, with the ALGCAR15 treatment presenting the greatest inhibition (22.08%) against this fungus. In contrast, when tests were carried out against *C. gloeosporioides* and *Penicillium* AT21H10, a lower inhibitory capacity was observed. In the confrontations carried out against spores of the *Penicillium* AT21H10 fungus, mostly low inhibition percentages were observed. Similar results were reported by Mohammadi et al. [1], who encapsulated lactic acid bacteria and obtained inhibitory effects of 22.5, 23.6, and 26.1% against spores of *Aspergillus niger*, *Aspergillus flavus,* and *Aspergillus parasiticus*, respectively.

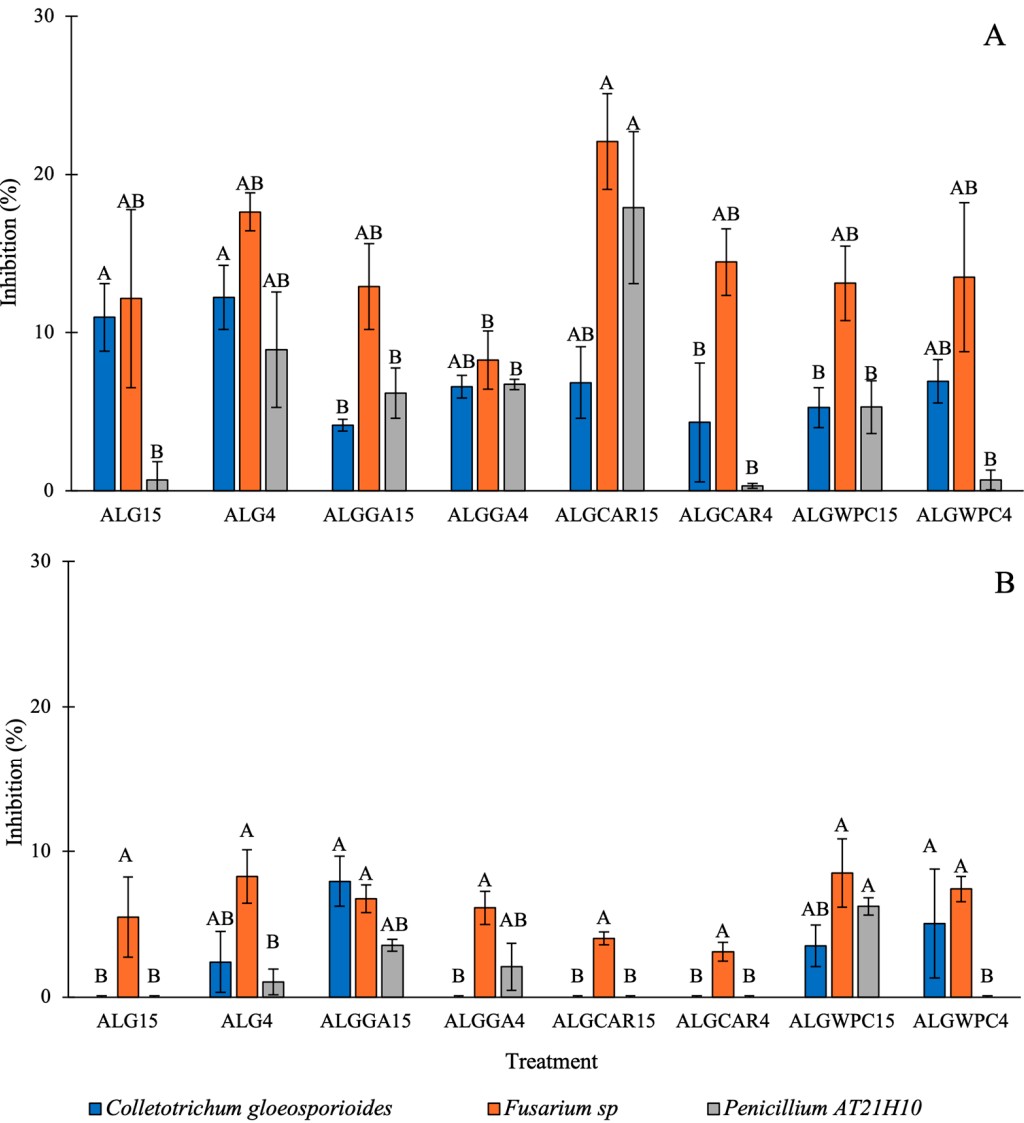

**Figure 5.** Inhibitory capacity of microencapsulated LAB against spores of phytopathogenic fungi. Fresh microcapsules (**A**) and after 14 days of storage at 4 °C (**B**). Different letters in bars of the same color indicate significant differences ($p < 0.05$) between the data according to Tukey's test. For details of the codes used for the treatments, see Table 1.

On day 14, a decrease in the inhibitory capacity of the encapsulated bacteria was observed (Figure 5B) with respect to the initial value. However, the same behavior observed on the initial day was preserved, since the confrontations carried out against *Fusarium* sp. continued to present higher inhibition values than those of the other phytopathogenic fungi studied. Although the freshly microencapsulated LAB inhibited the growth of the three fungi, this capacity was lost in three of the eight treatments by day 14. The microcapsules made of alginate and alginate–κ-carrageenan containing *L. plantarum* TEP15 did not inhibit the fungi *C. gloeosporioides* and *Penicillium* AT21H10. The same occurred with the ALGCAR4 treatment. This may be because the concentration of antimicrobial compounds within the capsules decreased over time or because they were not efficiently released into the culture medium, causing sufficient inhibitory activity to stop the germination of the fungal spores, unlike what occurred in the confrontations against mycelia. Spores are relatively stress-resistant structures and show great variation in their ability to survive under adverse conditions [65,66], which would help them survive or maintain their viability for longer and influence the effectiveness of treatments after 14 days of storage.

## 4. Conclusions

All the polymers used to microencapsulate the bacteria *Lactiplantibacillus plantarum* TEP15 and *Lactiplantibacillus pentosus* TEJ4 resulted in encapsulation rates greater than 86%, in addition to offering protection to the bacteria during 28 days of storage at 4 °C. No effect of the encapsulating polymers on the moisture content of the microcapsules was observed. The microcapsules made of alginate with gum arabic allowed us to obtain the smallest capsules (1.47–1.49 mm). The encapsulated LAB demonstrated greater inhibitory activity against mycelium than against spores of phytopathogenic fungi. In the confrontations against mycelia, the alginate treatment with *Lactiplantibacillus plantarum* TEP15 had the greatest inhibitory effect on *C. gloeosporioides* and *Penicillium* AT21H10, while the alginate treatment with κ-carrageenan containing *Lactiplantibacillus plantarum* TEP15 was effective against *Fusarium* sp., with inhibition values ranging from 37 to 100%. However, the highest values of spore germination inhibition reached 22.08% on day 0 and 8.51% on day 14. Additional investigations are required to better understand the antifungal dynamics of encapsulated lactic acid bacteria. Furthermore, it is necessary to test different encapsulating matrices and encapsulation techniques to increase the efficiency of these formulations for the biocontrol of phytopathogenic fungi.

**Author Contributions:** Conceptualization, Y.P.-R. and A.V.-O.; data curation, Y.P.-R.; formal analysis, Y.P.-R. and R.R.-Q.; investigation, D.G.-L.; methodology, Y.P.-R.; resources, M.S.-F.; supervision, D.M.-R.; validation, D.M.-R. and M.S.-F.; writing—original draft, Y.P.-R. and A.V.-O.; writing—review and editing, all authors. All authors have read and agreed to the published version of the manuscript.

**Funding:** This research received no external funding.

**Data Availability Statement:** Data are contained within the article.

**Conflicts of Interest:** The authors declare no conflicts of interest.

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
