# Peer review of "Antifungal Capacity of Microcapsules Containing Lactiplantibacillus plantarum TEP15 or Lactiplantibacillus pentosus TEJ4"

_processes, doi:10.3390/pr12040763_

Round 1

Reviewer 1 Report

Comments and Suggestions for Authors

Title: Antifungal Capacity of Microcapsules Containing Lactiplanti- bacillus plantarum TEP15 or Lactiplantibacillus pentosus TEJ4

Manuscript ID: processes-2926674

Dear Editor

The manuscript is about the encapsulation of two strains of TEP15 and TEJ4 and their antifungal activities. The manuscript gives use useful information. However, it needs major revision to improve its quality according to the comments:

·       Abstract: please insert the encapsulating material and their ration. Also, it needs to mention more results and details.

·       Introduction: please insert the encapsulating materials at the end of the introduction and provide the perospect for readers.

·       Material and methods, “microencapsulation” section: How did you chose the ratio of wall materials? Moreover, according to the Table 1, the dry matter of different formulations was not the same. The percentage of whey protein, carrageenan and gum Arabic were difference. How did you chose? And how can compared their effect when their percentage were different?

·       Table 2: please use upper statistical letters in table. Moreover, how were the statistical differences of samples during the storage time? For example, moisture content of ALG15 at day 28 was significantly higher that its moisture content at first day?

·       Microcapsule size, line 273-276: ALGGA 4 and ALGWPC4 had the smallest size and explained it was due to the viscosity of alginate. According to the Tabhe1, ALGGA had higher dry matter (2% alginate + 1% gum Arabic) and ALGWPC (0.75% alginate + 1.5% whey protein). Higher wall matter may cause higher bead size. Would you please  explain more and consider this aspect?

·       Survival rate: Please discuss in depth and explain more about the reactions between polymers and compared the interaction between polysaccharides- polysaccharides and polysaccharides-proteins.

·       Figure 3. The figure well exhibited the results. However, it should be contained the statistical differences between all treatments and each treatment between day 14 and 28.

·       How was the in vitro release of bacteria?  

Comments on the Quality of English Language

Title: Antifungal Capacity of Microcapsules Containing Lactiplanti- bacillus plantarum TEP15 or Lactiplantibacillus pentosus TEJ4

Manuscript ID: processes-2926674

Dear Editor

The manuscript is about the encapsulation of two strains of TEP15 and TEJ4 and their antifungal activities. The manuscript gives use useful information. However, it needs major revision to improve its quality according to the comments:

·       Abstract: please insert the encapsulating material and their ration. Also, it needs to mention more results and details.

·       Introduction: please insert the encapsulating materials at the end of the introduction and provide the perospect for readers.

·       Material and methods, “microencapsulation” section: How did you chose the ratio of wall materials? Moreover, according to the Table 1, the dry matter of different formulations was not the same. The percentage of whey protein, carrageenan and gum Arabic were difference. How did you chose? And how can compared their effect when their percentage were different?

·       Table 2: please use upper statistical letters in table. Moreover, how were the statistical differences of samples during the storage time? For example, moisture content of ALG15 at day 28 was significantly higher that its moisture content at first day?

·       Microcapsule size, line 273-276: ALGGA 4 and ALGWPC4 had the smallest size and explained it was due to the viscosity of alginate. According to the Tabhe1, ALGGA had higher dry matter (2% alginate + 1% gum Arabic) and ALGWPC (0.75% alginate + 1.5% whey protein). Higher wall matter may cause higher bead size. Would you please  explain more and consider this aspect?

·       Survival rate: Please discuss in depth and explain more about the reactions between polymers and compared the interaction between polysaccharides- polysaccharides and polysaccharides-proteins.

·       Figure 3. The figure well exhibited the results. However, it should be contained the statistical differences between all treatments and each treatment between day 14 and 28.

·       How was the in vitro release of bacteria?  

Reviewer 2 Report

Comments and Suggestions for Authors

The manuscript titled: ''Antifungal capacity of microcapsules containing Lactiplantibacillus plantarum TEP15 or Lactiplantibacillus pentosus TEJ4'' presents a study on the microencapsulation of lactic acid bacteria and the antifungal effect of microcapsules. Despite the interesting topic i have a few comments:  

Line 9-22: please list what polymers were used 

Line 35: without etc. please list the main ones 

Line 42: what is phytopathogenic fungi? 

Line 66: where did the changes in the reduction of antifungal activity come from? were there any trials conducted on the effect of changes regarding the effect of polymers? What were the thicknesses of the casing in the capsules? was the effect of the casing on the effect of antimicrobial activity studied? 

Line 75: How did the authors confirm that the isolated strains from fermented beverages and tejuino are tested? 

Line 93: On what basis did the authors conclude that this concentration is optimal? 

Line 106: Did the authors check how long the bacteria are alive after the encapsulation process? 

Line 96: please standardize names throughout the manuscript 

Line 111: Which bacteria were encapsulated in which experimental variant? 

Line 111: please explain the legend of abbreviations below the table 

Line 125: typo ‘’#4’’  

Line 181: What antimicrobial tests were conducted? 

Line 193: In how many repetitions were the experiments conducted? how many times were the bacteria isolated? How many replicates were the analyses performed? 

Line 219: explain the abbreviations under the table 

Line 289: are the statistics done correctly? if the errors are so large, and overlap the others then there should be no statistical differences 

Line 395: summary should only contain relevant information, please reformat conclusions

Comments on the Quality of English Language

Moderate editing of English language required

Round 2

Reviewer 1 Report

Comments and Suggestions for Authors

Title: Antifungal Capacity of Microcapsules Containing Lactiplanti- bacillus plantarum TEP15 or Lactiplantibacillus pentosus TEJ4

Manuscript ID: processes-2926674

Dear Editor

The manuscript is well improved, however, it needs some modifications before publication:

·       Please recheck Fig.5 (B) and modify.

·       Conclusion: Please insert some suggestions in conclusion for further studies.

Comments on the Quality of English Language

Title: Antifungal Capacity of Microcapsules Containing Lactiplanti- bacillus plantarum TEP15 or Lactiplantibacillus pentosus TEJ4

Manuscript ID: processes-2926674

Dear Editor

The manuscript is well improved, however, it needs some modifications before publication:

·       Please recheck Fig.5 (B) and modify.

·       Conclusion: Please insert some suggestions in conclusion for further studies.

Author Response

Dear reviewer,
We revised Figure 5B and corrected the errors of missing letters
We incorporate in the conclusion some suggestions for future research

We also carry out the language review with the help of a native English speaker.